# Unmasking the Mechanism behind Miltefosine: Revealing the Disruption of Intracellular Ca^2+^ Homeostasis as a Rational Therapeutic Target in Leishmaniasis and Chagas Disease

**DOI:** 10.3390/biom14040406

**Published:** 2024-03-27

**Authors:** Gustavo Benaim, Alberto Paniz-Mondolfi

**Affiliations:** 1Unidad de Señalización Celular y Bioquímica de Parásitos, Instituto de Estudios Avanzados (IDEA), Caracas 1080, Venezuela; 2Laboratorio de Biofísica, Instituto de Biología Experimental, Facultad de Ciencias, Universidad Central de Venezuela, Caracas 1040, Venezuela; 3Department of Pathology, Molecular and Cell-Based Medicine, Icahn School of Medicine at Mount Sinai, Division of Microbiology, New York, NY 10029, USA; alberto.paniz-mondolfi@mountsinai.org

**Keywords:** Ca^2+^ signaling, Ca^2+^ regulation, miltefosine, acidocalcisome, mitochondria, Ca^2+^ channel, sphingosine

## Abstract

Originally developed as a chemotherapeutic agent, miltefosine (hexadecylphosphocholine) is an inhibitor of phosphatidylcholine synthesis with proven antiparasitic effects. It is the only oral drug approved for the treatment of Leishmaniasis and American Trypanosomiasis (Chagas disease). Although its precise mechanisms are not yet fully understood, miltefosine exhibits broad-spectrum anti-parasitic effects primarily by disrupting the intracellular Ca^2+^ homeostasis of the parasites while sparing the human hosts. In addition to its inhibitory effects on phosphatidylcholine synthesis and cytochrome c oxidase, miltefosine has been found to affect the unique giant mitochondria and the acidocalcisomes of parasites. Both of these crucial organelles are involved in Ca^2+^ regulation. Furthermore, miltefosine has the ability to activate a specific parasite Ca^2+^ channel that responds to sphingosine, which is different to its L-type VGCC human ortholog. Here, we aimed to provide an overview of recent advancements of the anti-parasitic mechanisms of miltefosine. We also explored its multiple molecular targets and investigated how its pleiotropic effects translate into a rational therapeutic approach for patients afflicted by Leishmaniasis and American Trypanosomiasis. Notably, miltefosine’s therapeutic effect extends beyond its impact on the parasite to also positively affect the host’s immune system. These findings enhance our understanding on its multi-targeted mechanism of action. Overall, this review sheds light on the intricate molecular actions of miltefosine, highlighting its potential as a promising therapeutic option against these debilitating parasitic diseases.

## 1. Introduction

The hemoflagellate parasite *Leishmania* is the causative agent of leishmaniasis, which, depending on the infecting species, can generate three different clinical patterns: cutaneous, mucocutaneous, or visceral leishmaniasis, the latter being invariably mortal if left untreated. This parasitosis is one of the most neglected tropical diseases in the world and mainly affects the poorest countries, being the second cause of mortality worldwide, following malaria. Thus, with a mortality rate of 95% if left untreated, visceral leishmaniasis is responsible for approximately 50,000 to 90,000 new cases of infection worldwide each year, making it the second largest parasitic killer in the world after malaria [1].

Current treatment options for this disease includes the use of pentavalent antimonials (Glucantime^®^ and Pentostam^®^). However, these drugs have toxic effects in patients, and frequently, resistance is observed. In this sense, miltefosine, a drug that has been repurposed for the treatment of leishmaniasis, has been accepted as the only oral treatment approved for this disease, leading to a parasitological cure for visceral leishmaniasis [2].

On the other hand, a closely related organism, the parasite *Trypanosoma cruzi*, is the causative agent of American trypanosomiasis or Chagas Disease, a tropical human infection considered a neglected disease that causes significant mortality and morbidity in the whole Latin American subcontinent [3]. With globalization, this disease has begun to spread to North America and Europe. The development of new drugs against this infection is necessary since the current accepted therapy is based on the use of benznidazole and nifurtimox, with more than 50 years of use; although both drugs are partially effective in the acute phase of the infection, they have a very limited effect in the chronic phase (less than 25%), which is the most relevant and abundant form of this human infection [3]. Additionally, since their mechanism of action is based in the formation of free radicals, both drugs possess several adverse side effects, frequently leading to the interruption of the treatment [4].

Some repurposed drugs have been proposed for the treatment of Chagas disease, including the antiarrhythmic amiodarone [3] and its derivative dronedarone, and some other synthetized products based on amiodarone’s benzofuran structure [4]. Amiodarone and its derivatives have been shown to act in *Leishmania mexicana* and also in *Leishmania donovani*, the causative agents of visceral leishmaniasis. The mechanism of action of amiodarone and its derivatives in trypanosomatids has been recently reviewed [5].

Several inhibitors of the synthesis of ergosterol have also been proposed for the possible treatment of leishmaniasis and Chagas disease, since the parasites causing these infections possess this sterol structurally in its membranes (instead of cholesterol), which is of paramount importance for the survival of all trypanosomatids [6]. The polyene antibiotic amphotericin B offers an alternative treatment for visceral leishmaniasis, mainly in their liposomal presentation (AmBisome), with reduced toxicity and high effectiveness, but is also quite expensive. Other alternatives include paromomycin and pentamidine, which also display several side effects. Finally, miltefosine (hexadecylphosphocholine) is by far the most used treatment for diverse forms of Leishmaniasis, being at present the sole drug approved for oral administration. This compound was originally developed as an anti-cancer agent [7]. Indeed, miltefosine was used topically for metastases, but its lack of efficacy limited its use. However, this drug has proved to be very promising in the most fatal forms of leishmaniasis, including visceral leishmaniasis [8]. Nevertheless, the drug in not exempt from causing side effects. First, miltefosine is teratogenic, and should be avoided in infected pregnant women. Other minor side effects include gastrointestinal disturbance, vomiting, and diarrhea. On the other hand, it has been claimed to induce resistance; thus, its use is recommended in combination with other drugs, preferably with those inducing synergistic actions, such as amiodarone, as demonstrated using a mouse model [9]. In that study, it was shown, by using PCR as diagnostic method, that the combination of miltefosine with amiodarone produced a parasitological cure for 90% of treated infected mice. 

Interestingly, the main mechanism of action of miltefosine appears to be through the disruption of the parasite’s intracellular Ca^2+^ homeostasis [10], as this drug affects several organelles and channels directly involved in Ca^2+^ regulation. The same case results for the interaction of this drug on *T. cruzi* [11]. In addition to miltefosine, several other drugs have been reported to affect the intracellular Ca^2+^ concentration ([Ca^2+^]_i_), indicating that the latter is the main mechanism of action of numerous drugs [12], including amiodarone [3] and its derivatives [5]. Accordingly, it has been stated that affecting the parasite’s Ca^2+^ homoeostasis could be considered a vital target of choice when looking for the effect of different drugs on trypanosomatids [12,13]. Taking this into consideration, it is worth dedicating a detailed description of the main mechanisms involved in Ca^2+^ regulation in trypanosomatids compared to the human or mammal counterpart, which appears to be quite different in many aspects. 

## 2. Intracellular Ca^2+^ Regulation in Trypanosomatids and Comparison with Mammal Cells

Ca^2+^ is an essential messenger in many fundamental functions in all eukaryotes, and the [Ca^2+^]_i_ should be finely regulated. This is especially important when taking into consideration the existence of a wide range of difference between the cytoplasm Ca^2+^ concentration (50–100 nanomolar in resting cells) and the extracellular *milieu* (around 2 mM). Indeed, this is warranted by the presence of different transport mechanisms, some of them located in intracellular organelles and others at the plasma membrane [14]. With respect to intracellular organelles, mitochondria possess in their internal membranes an electrophoretic Ca^2+^ uniporter (MCU) which utilizes, as the driving force for Ca^2+^ accumulation, the difference in the electrochemical potential (Δφ_m_) generated by the translocation of H^+^ from the intramitochondrial space to the cytoplasm. This mechanism allows for the accumulation of large amounts of calcium, which can leave the mitochondria through an electroneutral Ca^2+^/H^+^ (or a Ca^2+^/Na^+^) exchanger (Figure 1). This mechanism has been studied and well characterized [14], and appears to be well conserved through evolution [13,15]. Mitochondrial Ca^2+^-transport in different species of trypanosomatids has been demonstrated in whole cells by the use of digitonin-permeabilized cells in *T. cruzi* [16]. This technique has proven to be very useful because it allows the contribution of different intracellular compartments to ionic homeostasis to be dissected. At nearly the same time, using a more direct approach by means of coupled mitochondrial vesicles obtained from *L. braziliensis*, it was also demonstrated that this organelle shares most of the essential properties present in its mammal counterpart [17]. Shortly after these publications, the involvement of mitochondrion in Ca^2+^ regulation was well documented in all trypanosomatids studied so far, supporting the notion that the presence of an MCU and other features of the mitochondrial function are well conserved through evolution [13,15]. However, and very different to most eukaryotic cells, including mammals, trypanosomatids possess a single unique mitochondrion that occupies around 12% of the whole cytoplasm (Figure 2). In fact, the mitochondrial DNA (kinetoplast) is so huge that it is clearly visible with the help of a light microscope, thus giving the name to the whole family: Kinetoplastidiae [13].

Similar to mammal cells, the endoplasmic reticulum from trypanosomatids is also involved in Ca^2+^ regulation, as was demonstrated in experiments with digitonin-permeabilized *L. donovani* [18]. However, this organelle in trypanosomatids is not sensitive to the signaling molecule inositol 1,4,5-trisphosphate (IP_3_) [19], pointing out that the endoplasmic reticulum is less important concerning the [Ca^2+^]_i_ regulation in these parasites. Indeed, this signaling pathway has been substituted in these parasites by another fundamental organelle with an essential function not only in Ca^2+^ regulation, but in the bioenergetics of the entire Kinetoplastidiae family, namely, the acidocalcisomes. This organelle, first described in trypanosomes [20,21], has expanded its relevance, since many functions appear to involve their purpose in trypanosomatids as well as in other human parasites from the Apicomplexa family, causing malaria and toxoplasmosis [22]. Acidocalcisomes constitute the largest reservoir of intracellular Ca^2+^, even with more relative capacity when comparing this property with the giant mitochondrion mentioned above. Accordingly, this organelle also possesses a Ca^2+^-ATPase for Ca^2+^ uptake and accumulation. Besides calcium, acidocalcisomes are also enriched in other inorganic cations such as magnesium, sodium potassium, and zinc. Also, they are acidic organelles, being acidified by a vacuolar-type proton-translocating (V-H^+^-ATPase) pump. Acidocalcisomes are, in various ways, similar to the vacuole present in fungi and plant cells, but have no counterpart in mammal cells [22,23]. Acidocalcisomes undoubtedly deserve more investigation due to their function in intracellular Ca^2+^ homeostasis, their direct relation with infectivity of the parasite, and their connection with pH regulation and with osmoregulation, among other important roles in these parasites [23]. One of the main facts that directly associates acidocalcisomes with intracellular Ca^2+^ homeostasis is the remarkable evidence that the IP_3_ receptor, instead of its ubiquitous presence at the endoplasmic reticulum, as it occurs in all mammal cells, is localized at the acidocalcisome membrane in trypanosomatids [24]. Interestingly, as recently reported in *T. cruzi*, the IP_3_ receptor-mediated Ca^2+^ release from acidocalcisomes has been directly connected to the regulation of mitochondrial bioenergetics, also preventing parasite autophagy [25]. Acidocalcisomes provide another paramount function, since they are also capable of accumulating large quantities of phosphorus, present as phosphates, pyrophosphates, and polyphosphates, together with an enzyme with pyrophosphatase activity. This fact is relevant for parasite bioenergetics since it has been shown that pyrophosphate constitutes an alternative energy source to ATP, and is frequently used in many transporting functions in trypanosomatids [23]. A proton-pumping pyrophosphatase similar to the present one in acidocalcisomes has also been localized and characterized at the plasma membrane and at the Golgi apparatus of *T. cruzi*, thus revealing its ubiquity in these parasites [26]. The role of acidocalcisomes has been extensively studied, mainly in trypanosomatids. These parasites are submitted to drastic fluctuation in osmolarity during various stages along their life cycles. For example, in the insect vector lower digestive track, epimastigotes of *T. cruzi* are submitted to dramatic oscillation in osmolarity that reaches a value of up to 1000 mosmol/kg in its yellow track liquid. Then, when transformed to blood stages, the osmolarity returns to 300 mosmol/kg [25]. Additionally, these parasites need to regulate their volume constantly. In fact, it has been shown that acidocalcisomes can fuse with the parasite contractile vacuole [23]. Other relevant functions of acidocalcisomes involve the regulation of the parasite’s pH homeostasis, their participation in autophagy, and their role in infectivity [23]. In the search for new drugs against the several infections produced by these parasites, all the relevant roles mentioned above, together with the large differences with the host cells, pave the way for focusing on acidocalcisomes as a rational, direct target [23,25]. In fact, as we will see below, this organelle is one of the main targets of miltefosine, in accordance with the notion that disruption of intracellular Ca^2+^ regulation is lethal for trypanosomatids. 

Concerning the intracellular Ca^2+^ regulation, the calcium-transporting systems present in intracellular organelles can regulate intracellular Ca^2+^ quite efficiently for short-term periods. However, for long-term homeostasis, they are limited by their capacity as intracellular compartments. Thus, the long-term regulation depends on the transport mechanisms positioned in the plasma membrane, where they can pump out Ca^2+^ without any volume restriction. The presence of a plasma membrane, Ca^2+^-ATPase (PMCA), in trypanosomatids was first demonstrated in *L. braziliensis* many years ago using an enriched plasma membrane fraction. Like its human counterpart, the enzyme is stimulated by calmodulin, which increases its affinity for Ca^2+^ to a level compatible with its very low concentration of around 50 nanomolars [13], therefore being able to pump Ca^2+^ to the outside *milieu* against a large gradient of about four orders of magnitude [27]. After this pioneer study, and using the same general protocol, the PMCA was further characterized and even purified by means of a calmodulin affinity column in other trypanosomatids, such as *T. cruzi* [12], *L. mexicana* [5], and *T. brucei* [28]. The affinity for Ca^2+^ found in all these parasite pumps in the presence of calmodulin is compatible with their function as responsible for the fine-tuning of the basal cytoplasmic Ca^2+^ concentration, which is essential for the appropriated function of Ca^2+^ as a second messenger. 

Interestingly, in the case of the Ca^2+^ pump from *T. brucei*, and supporting the notion that the disruption of Ca^2+^ homeostasis is key for a target search for drugs that affect trypanosomatids, it has been demonstrated that pentamidine, a drug approved for the treatment of sleeping sickness caused by this parasite, and also for the treatment of certain cases of leishmaniosis, is able to inhibit, by direct interaction with the enzyme, the ATPase activity and the associated Ca^2+^ transport of the parasite Ca^2+^ pump without producing any discernible effect on the ortholog PMCA from humans [28]. Instead, on the human PMCA, pentamidine behaves as a poor calmodulin (CaM) antagonist [28]. In the same context, crystal violet, a dye described as effective against *T. cruzi* trypomastigotes in the blood, was demonstrated to exert its anti-trypanosoma effect by disruption of the parasite’s Ca^2+^ homeostasis, including inhibition of the Ca^2+^-ATPase activity and associated Ca^2+^ transport, dissipation of mitochondrial electrochemical membrane potential, and inducing Ca^2+^ release from the endoplasmic reticulum [29]. 

Regarding Ca^2+^ regulation and function, CaM deserves special attention. This protein is considered the most ubiquitous and important Ca^2+^-binding protein, able to undergo multiple post-transcriptional modification [30], which notably confers the protein extreme plasticity, thus increasing its ability to recognize many distinct enzyme targets. Being an intracellular Ca^2+^ sensor thanks to presence of four high-affinity Ca^2+^-binding domains showing highly cooperative behavior, this protein is directly involved in decoding Ca^2+^ signaling. CaM has been extensively studied, and it has been shown that it possesses 148 amino acids with very particular composition. Remarkably, 50 amino acids are acidic (aspartate and glutamate), conferring this protein a low Pka (about 4). The protein also lacks cysteine, conferring larger flexibility. This protein has been very well conserved throughout evolution, being identical in all vertebrates. CaM has been cloned, sequenced, and characterized in different trypanosomatids [13,31], showing interesting differences with respect to vertebrates. Calmodulin from trypanosomatids has 148 amino acids and shows 17 substitutions, being an exception among eukaryotes. This fact should imply possible different mechanisms of interaction with the target proteins. This seems to be the case of the trypanosomatids PMCA described above and in other closely related trypanosomatid *T. equiperdum*, since the aminoacidic sequence does not show the characteristic CaM-binding domain found in the orthologue enzyme from mammals [32]. It is plausible that trypanosomatids could possess possible different unknown targets for this Ca^2+^-modulated protein, thus deserving further investigation regarding its function in these parasites. We performed a theoretical calculation of CaM concentration in *L. braziliensis* based on the volume of promastigotes and the amount of CaM obtained from a known promastigote sample. This calculation gave an amazing concentration of 20 µM, like the values obtained in brain and rat testis, which are considered organs with higher concentrations of this protein in mammals [14]. This concentration would account for the binding of 80 µM of Ca^2+^, taking into consideration that CaM possesses four Ca^2+^ high-affinity binding sites, indicating a potential large buffer capacity of the protein in these parasites. In any case, the huge amount of CaM in these parasites would suggest that this pivotal Ca^2+^-binding protein could be responsible for an unknown function in trypanosomatids. 

The presence of a Ca^2+^/Na^+^ exchange mechanism was evaluated at the plasma membrane of these parasites. Thus, experiments were performed using the plasma membrane vesicles mentioned above for the characterization of the PMCA obtained from *T. cruzi* [12], *L. mexicana* [5], and *T. brucei* [28]. In these experiments, the involvement of a Ca^2+^/Na^+^ exchanger at the plasma membrane of these parasites was ruled out. Thus, the addition of ATP, which allows for the accumulation of Ca^2+^ inside the vesicles, did not induce the release of Ca^2+^ to the extravesicular medium. This, however, is not surprising, since the plasma membrane Ca^2+^/Na^+^ exchanger as a Ca^2+^ extrusion mechanism is less ubiquitous than the Ca^2+^ pump, being more limited to excitable cells. This also indicates that the long-term intracellular Ca^2+^-regulation in these parasites resides, so far, exclusively on the plasma membrane Ca^2+^-ATPase.

Finally, the mechanism for Ca^2+^ entry in trypanosomatids has been more recently elucidated. A Ca^2+^ channel has been described at the plasma membrane of *L. mexicana* [32], and more recently on *T. cruzi* [12]. This channel is an analog of the human L-type voltage-gated Ca^2+^ channel (L-type VGCC). Hence, it is inhibited by the classical L-Type VGCC antagonist nifedipine (a dihydropiridine) and verapamil (a phenylalkylamine) and stimulated by the human L-Type VGCC agonist, Bay K 8644. These characteristics agree with well-conserved sequences among the predicted parasite channels and that from the human L-Type VGCC [11,32]. However, there are remarkable differences, mainly evidenced by the electrophysiological characterization of the *T. cruzi* channel, when compared to the human L-type VGCC. For example, the latter has a marked voltage dependency, which triggers its opening after an action potential, inducing a Ca^2+^ entrance to the cell, while the parasite channel appears not to be voltage-dependent [11]. Another outstanding difference with respect to the human counterpart is that the parasite channel is specifically stimulated by the sphingolipid sphingosine (Sph), while the human channel is not. Interestingly, the parasite channel is also a miltefosine target [10,11] and resembles the effect of the natural sphingolipid, not only in its interaction with the Ca^2+^ channel but also in its molecular structure, as can be seen in Figure 1. This is also the case of the Ca^2+^ channel recently reported for *T. equiperdum* [33], a trypanosomatid that infects horses, which is somehow different to the other channels found in human parasites described above. This will be discussed in the next section, which is focused on the mechanism of action of this drug on trypanosomatids.

## 3. Mechanism of Action of Miltefosine

Miltefosine is a pleiotropic drug with multiple targets in different trypanosomatids and mammal cells [10], including diverse positive effects on the immune system of the infected host, apparently also responsible for its beneficial action [34]. This unusual combination of desirable effects is, in all likelihood, the reason for the success of this drug as oral monotherapy against leishmaniasis. For the above reason, in this review, we will attend separately to these two important properties. 

## 4. Mechanism of Action of Miltefosine in Trypanosomatids 

The first mechanism of action demonstrated for miltefosine on trypanosomatids is related to the impairment of phospholipids biosynthesis, inducing a significant decrease in several phospholipids and amino acids, while simultaneously increasing sphingolipids and sterols. Specifically, miltefosine alters the phospholipid content in *L. donovani* and *L. major*, halving the concentration of phosphatidylcholine while doubling that of phosphatidylethanolamine compared to the control [34]. Lysophospholipids generally induce a pronounced effect on the phospholipid composition of trypanosomatids. In *Trypanosoma cruzi*, miltefosine inhibits the Greenberg (transmethylation) pathway by acting on phosphatidylethanolamine N methyl-transferase [35,36]. Conversely, miltefosine inhibits phosphatidylcholine biosynthesis in mammal cells, primarily via the Kennedy CDP–Choline pathway, by blocking phosphocholine citidyltransferase [36]. Interestingly, it has been suggested that miltefosine inhibits phosphatidylcholine biosynthesis in *T. cruzi* with 10 to 20 times more potency than in mammalian cells, thus explaining its high selectivity as an antiparasitic drug [36]. In the case of *L. donovani*, a similar mechanism has been demonstrated, with a decrease in phosphatidylcholine concentration and an increase in phosphatidylethanolamine concentration [37]. According to Rakotomanga et al. 2007, the reduction in phosphatidylcholine in this parasite, both total and from plasma membranes, could be attributed to miltefosine’s inhibition of phospholipid synthesis via the CDP–choline pathways, affecting CTP-phosphocholinecytidylyltransferase activity. Additionally, phosphatidylcholine synthesis via the CDP–choline pathway requires transport of the choline precursor from the host, which is also inhibited by miltefosine in *L. major* promastigotes [37]. Furthermore, it has been demonstrated across various cell lines, including tumor cells, that miltefosine inhibits exogenous choline incorporation into phosphatidylcholine (de novo synthesis) due to a modification in CTP-phosphocholine-cytidylyltransferase activity.

Miltefosine also induces an increase in the amount of cholesterol to the double in *L. donovani* membranes [38]. Since this sterol is not a product of de novo sterol biosynthesis by these parasites, it should be taken from the external milieu. It has been hypothesized that miltefosine stimulates cholesterol uptake via a condensation effect between the drug and sterols [38]. Concerning trypanosomatids sterols, an increase in the levels of ergosterol induced by miltefosine treatment has been demonstrated in *L. donovani* the level of sphingolipids. Specifically, Sph was found to be elevated in *L. donovani* and *L. major* using axenic amastigotes by the use of a novel method developed to study the metabolomes and lipidomes of these parasites [39]. In this same system, Armitage et al., 2018, indicated a dramatic fivefold increase in sphingosine content. Ceramide levels were also elevated upon miltefosine treatment. The authors suggest that the increase in ergosterol in *L. donovani* and *L. major*, coordinated with an increase in sphingolipids, could point toward a complex interplay, and postulate that the function of lipid microdomain complexes of sterols and sphingolipids may possibly be involved in the mechanism of action of the drug on these parasites. Taken together, it has been suggested that the exceptional lipid composition (phospholipids, sphingolipids, and ergosterol) renders *Leishmania* parasites more resistant to lipid perturbation that is generally fatal to other organisms [40].

Another effect of miltefosine in trypanosomatids is related to the induction of an apoptosis-like process, thus leading to parasitic death. For example, in *L. donovani*, it has been demonstrated that miltefosine induces nuclear DNA condensation and DNA fragmentation, including ladder formation and in situ labeling of DNA, by the TUNEL method [41]. The concept of apoptosis in unicellular organisms has sparked considerable debate regarding the evolution of apoptotic mechanisms in individual cells. This “altruistic” behavior might seem unexpected, as apoptosis is traditionally viewed as a mechanism for purging undesirable cells in multicellular organisms [42,43]. However, considering population survival as an objective, sacrificing some individual parasites to benefit the host’s survival makes sense. If the infection becomes too severe, the host may perish, thereby halting the transmission of the infection to new hosts—a significant evolutionary disadvantage. Nevertheless, miltefosine-induced apoptotic-like mechanisms have been extensively documented in various reports on trypanosomatids, encompassing a wide array of apoptotic parameters [42,43,44,45,46,47]. Among these features, the translocation of phosphatidylserine (PS) to the outer leaflet of the plasma membrane bilayer, as demonstrated in *T. cruzi* [46] and *Leishmania* [47], warrants special consideration. This exposure of PS serves as a signal for apoptotic cells to be engulfed by macrophages without triggering an undesirable inflammatory response from the immune system—a natural mechanism for eliminating apoptotic residues. In the case of these parasites, PS exposure prompts macrophages to engulf them while simultaneously evading the inflammatory response through the production of anti-inflammatory cytokines such as TNFβ [45,48]. This “Trojan horse” strategy is considered a co-evolutionary adaptation in trypanosomatids, facilitating safe entry into macrophages and favoring the survival of intracellular parasites. This mimicry also involves inhibiting nitric oxide production within the phagocyte by down-regulating iNOS production [45].

The specific mechanism of induction of apoptosis by miltefosine is not known, but since apoptosis is a well-known Ca^2+^ mediated process and involves Ca^2+^ signaling in several fragments of this complex progression [49], it is conceivable that, since miltefosine has a profound effect on Ca^2+^ regulation in trypanosomatids (see below), part of its mechanism of action could be mediated by the disruption of Ca^2+^ homeostasis in the parasite.

Another essential target concerning the mechanism of action of miltefosine is the large mitochondrion present in trypanosomatids. In a pivotal report, it has been demonstrated that miltefosine causes a decrease in the oxygen consumption rate, consequently affecting ATP levels in *L. donovani* through the inhibition of the mitochondrial cytochrome C oxidase [50]. A dose-dependent effect of miltefosine on the mitochondrial electrochemical potential (Δφ_m_) of L. donovani was also observed by the use of Rhodamine 123 as a fluorescence indicator with a maximal effect observed at 40 µM miltefosine, but with no discernible effect at 10 µM. However, using a different experimental approach, a rapid and direct effect on the depolarization of Δφ_m_ with 4 µM miltefosine, a concentration one order of magnitude lower, has been reported [10]. The difference in the experimental procedure was essentially that, in the previous work [50], the parasites were incubated with the drug for 14 h, then washed, and the fluorescence of rhodamine 123 was determined using a flow cytometer [50], while in the more recent work [10], miltefosine was added directly and stirred in a thermostated cuvette in a fluorimeter, and rhodamine 123’s fluorescence was followed in real time. Under this condition, a rapid collapse of the Δφ_m_ was observed within seconds, reaching a maximal value in 5 min (Figure 3A). The results obtained with this protocol resemble the effects of other compounds postulated to act as protonophores on the mitochondria. This is the case of the antituberculosis drug SQ109, which has been shown to produce an effect in the mitochondria of other trypanosomatids, as is the case of *T. cruzi*. The experiments showed a rapid collapse of the inner mitochondrial membrane potential Δφ_m_, dissipating the proton gradient by acting as the uncoupler or protonophore FCCP [6]. The same result has been found in *L. mexicana* [51], and also in *L. donovani* [7].

In support of this proposed mechanism of action of miltefosine on the mitochondrion, other drugs mentioned previously, such as amiodarone [3] and dronedarone [4], are known to affect the mitochondria of *T. cruzi* [4], following the same general pattern. Even more, this was also the case of amiodarone and dronedarone when studied on *L. mexicana* mitochondrion [5]. In conclusion, the similarities of the effect of miltefosine on the mitochondrion when compared to all the mentioned drugs allow us to suggest that this drug, besides producing a marked inhibition of cytochrome c oxidase, also has protonophoretic action, rapidly collapsing the Δφ_m_. This will generate a calcium efflux, increasing the cytoplasmic Ca^2+^ concentration and its consequences for the parasite’s survival, as will be explained in the following section.

## 5. Intracellular Ca^2+^ Homeostasis as a Target for Miltefosine Mechanism of Action in Trypanosomatids

As mentioned previously, several drugs exert their antiparasitic effects through the disruption of intracellular Ca^2+^ regulation. Miltofosine is not an exception. On the contrary, it is a drug with more different actions on [Ca^2+^]_i_ regulation than those described so far in these parasites. Thus, this drug affects the unique giant mitochondria and the acidocalcisomes, both organelles devoted to Ca^2+^ regulation and also directly involved in the parasite’s bioenergetics, as mentioned previously. Furthermore, and crucial for understanding its mechanism of action, miltefosine has the ability to activate a specific parasite Ca^2+^ channel that responds to sphingosine (Figure 4). To our knowledge, this is the only drug that elevates the [Ca^2+^]_i_ inducing its release from intracellular organelles, and simultaneously allows for the entrance of Ca^2+^ from the extracellular *milieu* by activating the Sph-activated-Ca^2+^ channel [12]. The combination of these different targets produces an elevated increase in the parasite’s [Ca^2+^]_i_ regulation, leading to its death.

With regard to the release of Ca^2+^ from the mitochondrion, since the driving force for the Ca^2+^ entrance is the mitochondrial membrane potential Δφ_m_, the collapse caused by the drug action produces the release of the accumulated Ca^2+^ from the mitochondrion to the cytoplasm. On the other hand, miltefosine directly leads to the alkalinization of the acidocalcisomes (Figure 3B), another essential reservoir of Ca^2+^, from which this cation is rapidly released to the parasite cytoplasm [12]. Concerning intracellular organelles, it is well known that acidocalcisomes and the mitochondrion can accumulate large quantities of Ca^2+^, as was observed many years ago upon the addition of the Ca^2+^ ionophore A-23187 to entire parasites in the presence of digitonin, allowing for entrance of the Ca^2+^ ionophore to the intracellular organelles. Under this condition, a massive release of Ca^2+^ was observed [17]. It is not difficult to conceive that, after miltefosine uptake by the infected pacient, the simultaneous release of Ca^2+^ from intracellular organelles, and the entrance of the cation from outside, would produce a massive increment in the [Ca^2+^]_i_ that cannot be managed by the homeostatic [Ca^2+^]_i_ mechanism, nevertheless being very efficient, and thus producing the parasite death.

Regarding the interaction of miltefosine with the Sph-sensitive Ca^2+^ channel, as shown in Figure 5 and Table 1, there is a clear similarity between sphingosine and miltefosine. Thus, it is conceivable that this drug replaces the sphingolipid upon its interaction with the channel, as previously suggested [11]. It could be argued that de novo synthesis of Sph is behind the mechanism of action of miltefosine, as an action of the drug has been demonstrated on the metabolome of *Leishmania*, leading to Sph synthesis [40]. However, the possible activation of the Ca^2+^ channel via Sph de novo synthesis was ruled out, since miltefosine’s effect was observed using “patch clamp” techniques on plasma membranes vesicles fused to giant liposomes [11] that were devoid of the machinery for the de novo synthesis of sphingolipids. Thus, a direct effect of miltefosine on this Ca^2+^ channel was demonstrated.

We analyzed the ADME parameters of sphingosine and compared them with miltefosine in order to gain a deeper understanding of the possible similar patterns that can be recognized by the same binding site of a biological target and, therefore, exhibit similar biological profiles. The ADME parameters were determined using the tools available on the SWISSADME server [52]. Table 1 summarizes the six physicochemical parameter derivatives from Lipinski’s “rule of five”—lipophilicity, size, polarity, solubility, saturation, and flexibility—in the radar representations for the compounds [52,53]. Interestingly, both compounds exhibited similar patterns in terms of the properties and characteristics. To be estimated as drug-like, the red line of the compound under study must be fully included in the pink area. For lipophilicity, XLOGP329 should be in the range from −0.7 to +6.0; miltefosine has a higher lipophilicity. For flexibility, the molecule should not have more than nine rotatable bonds. Sphingosine and miltefosine are both highly flexible, with 15 and 20 rotatable bonds, respectively.

With respect to the Sph-sensitive Ca^2+^ channel, it is interesting to note that the human ortholog, the L-type VGCC (or CaV1.2), is extremely regulated by CaM. In fact, CaM is able to directly interact with three different domains of human Ca^2+^ channel, conferring, among other properties, Ca^2+^ inactivation [54]. Taking into consideration that the trypanosomatid Ca^2+^ channel possesses IQ domains [33] that are characteristic of CaM-binding domains, and that trypanopsomatids possess large quantities of CaM with an unique amino acidic sequence with 17 substitutions, it would be interesting to study the effect of CaM in relation to the effect of miltefosine and sphingosine on this Ca^2+^ channel.

## 6. Effects of Miltefosine on the Host Immune System

Leishmaniasis presents a complex interplay of factors influencing disease progression, with various cell types and immune responses playing critical roles. Here, we explore the multifaceted dynamics between *Leishmania* parasites and the host immune system, shedding light on the potential impact of repurposing miltefosine. From the involvement of innate immune cells like neutrophils and macrophages to the modulation of cytokine responses and the intricate balance of Th1 and Th2 immunity, we delve into the complexities underlying leishmanial infection. Moreover, we discuss the immunomodulatory effects of miltefosine on different immune cell populations, highlighting its potential implications for disease pathogenesis and treatment outcomes. Through a comprehensive analysis of these interactions, we aim to contribute to the development of more effective therapeutic strategies against this challenging parasitic disease.

Over time, extensive research has deepened our understanding of the dynamics between *Leishmania* parasites and their mammalian hosts, as well as the complex immune responses that govern this interaction. Here, we explore how various cell types contribute to initiating and modulating the immune defense against the infection, along with examining the potential immunomodulatory effects of miltefosine.

The interaction of diverse *Leishmania* parasite species is known to modulate multiple innate immune cells, altering their characteristics and functions and influencing adaptive immune responses [55]. Shortly after the parasite gains access to the host via inoculation through sand fly saliva, a cascade of inflammatory events is triggered. This includes the recruitment and activation of innate cell populations such as mast cells, neutrophils, eosinophils, macrophages, and monocytes, which later evolve into dendritic cells (DCs) once recruited into the lesional *milieu* (Figure 6). Neutrophils, macrophages, and DCs all play a dual role, capable of either eliminating or supporting parasite survival [55]. Miltefosine has demonstrated significant immunomodulating effects on these cells (Figure 6), a topic we will examine in detail.

Neutrophils recruited to the infection site eliminate *Leishmania* parasites through phagocytosis, the release of reactive oxygen species (ROS), and neutrophil extracellular traps (NETs) [56,57]. However, it is widely known that *Leishmania* can transiently survive within neutrophils by hindering phagolysosome formation, suppressing oxidative stress, and delaying neutrophil apoptosis. A study by Mukhopadhyay et al. [58] showed that miltefosine increased the secretion of pro-inflammatory cytokines, particularly the neutrophilic chemokine IL-8 (CXCL8), which, in turn, attracts more neutrophils and phagocytic cells, thus aiding *Leishmania* survival and pathology. On the other hand, a study performed by Regli et al. on both human and mouse neutrophils showcased interesting results, observing that neutrophil phenotype activation and functionality relied on the drug susceptibility profile [59]. The study demonstrated that in vitro infections with antimonial or miltefosine-resistant *Leishmania* (*Viannia*) *panamensis* parasite lines revealed increased neutrophil functions, such as enhanced reactive oxygen species (ROS) production and neutrophil extracellular trap (NET) formation [59]. The study showed that drug-resistant parasites induce higher neutrophil activation, but paradoxically show increased survival within neutrophils compared to susceptible strains [59]. These findings suggest a complex interplay between drug susceptibility, neutrophil activation, and *Leishmania* survival (Figure 6), shedding light on factors influencing susceptibility to miltefosine, with potential implications for miltefosine resistance in *Leishmania* [59].

However, neutrophils are not the only cell population from the granulocyte lineage that are susceptible to being influenced by the pleotropic effects of miltefosine. Miltefosine has been proven to suppress the activation of human eosinophils, hence leading to a decreased inflammatory response [60] that can ameliorate tissue destruction and, hence, the severity of symptoms. Miltefosine induces IL-12-dependent Th1 responses, crucial for combating *Leishmania* infections by bolstering cellular immune responses [61,62]. In fact, a study by Das et al. demonstrated that a combination therapy with paromomomycin/miltefosine has the ability to modulate TLR9 on dendritic cells, hence impacting Th1 immune responses in leishmaniasis therapy [63] A study by Knuplez et al. recently showed that miltefosine suppresses eosinophilic effector reactions, including CD11b up-regulation, degranulation, chemotaxis, and downstream signaling [61]. In their study using mouse models, the authors demonstrated that miltefosine significantly curtails immune cell infiltration in the respiratory tract during allergic cell recruitment and enhances lung function parameters in allergic inflammation [60] These findings indicate a robust modulatory role of miltefosine in regulating eosinophilic inflammation both in vitro and in vivo [61]. These findings reveal the potential utility of miltefosine in treating not only allergic diseases and other eosinophil-associated disorders, but also potential immunomodulatory effects, in patients affected by *Leishmania* [60]. 

Among the various granulocytes, mast cells, being tissue-resident, play a significant role in *Leishmania* infection [63,64]. Encountered early by invading pathogens, these cells function as effectors in innate immunity. Their abundance in the superficial dermis, where *Leishmania* is commonly encountered in the cutaneous niche, suggests their crucial involvement in immune responses to *Leishmania*. In a study conducted by Naqvi et al. [64], the interaction between *L. donovani*, the causative agent of visceral leishmaniasis, and *L. tropica*, responsible for both visceral and cutaneous leishmaniasis, with mast cells (MCs) played a substantial role in disease pathogenesis. The authors observed that co-culturing *Leishmania* with peritoneal mast cells (PMCs) from BALB/c mice and rat basophilic leukemia (RBL-2H3) MCs led to significant killing of *L. tropica* and, to a lesser extent, of *L. donovani* [65]. Notably, there were differences in phagocytic activity, with MCs exhibiting substantial uptake of *L. tropica* but no phagocytosis of *L. donovani* [66]. Additionally, a significant generation of ROS by MCs upon co-culture with these *Leishmania* species was observed, potentially contributing to their clearance. Furthermore, MC–*Leishmania* interactions resulted in the formation of extracellular traps comprising DNA, histones, and tryptase, serving to ensnare these pathogens [65]. This study conclusively established that MCs may contribute differentially to host defenses against *Leishmania* by actively internalizing these pathogens and deploying effector responses for their extracellular clearance [65].

Miltefosine, a known inhibitor of degranulation, microtubule reorganization, and antigen-induced chemotaxis in mast cells, operates not only through lipid raft modulation, but also by inhibiting Ca^2+^-dependent PKCs, influencing cytosolic signaling pathways governing microtubule organization, degranulation, and mast cell migration [66]. Additionally, research indicates that *Leishmania donovani* extracts membrane cholesterol from macrophages, disrupting lipid rafts and impairing their ability to stimulate T cells [67]. This suggests that miltefosine might play a potential role in modulating disease pathogenesis, possibly by altering lipid rafts and preventing phagolysosomal evasion, a phenomenon also described for *T. cruzi* [68]. 

Various cells, cytokines, and immune mediators [69,70], such as interleukins, IFNγ (interferon-gamma) [70], ROS [71], NO (nitric oxide) [72], NETs (neutrophil extracellular traps) [73], MIP (macrophage inflammatory protein) [74], TNFα [75], TGFβ (transforming growth factor-beta) [74,76], and NK (natural killer) [77], significantly contribute to immune responses against the *Leishmania* parasite. For example, in a study by Mukhopadhyay et al. [58] investigating the immunomodulatory effects of miltefosine on cytokine expression in post-kala-azar dermal leishmaniasis (PKDL), the authors assessed the drug’s immunomodulatory activity by examining activation markers such as CD14 and CD16, as well as the costimulatory molecules CD80 and CD86, on circulating monocytes. Additionally, they monitored the levels of pro-inflammatory cytokines (TNF-α, IL-6, IL-1β, and IL-8) and anti-inflammatory cytokines (IL-10, TGF-β, IL-4, and IL-13) in both serum and peripheral blood mononuclear cell culture supernatants. The results from this study revealed that miltefosine upregulated CD16 and CD86 expression on circulating monocytes while reducing CD14 expression, suggesting that the drug promotes a pro-inflammatory phenotype in monocytes, leading to heightened secretion of pro-inflammatory cytokines. Furthermore, the serum nitrite and arginase activity were monitored, showing that miltefosine also displayed macrophage-activating potential by decreasing serum arginase activity and increasing serum nitrite levels [58].

Natural killer cells are another subset of cells that exhibit a protective role in leishmaniasis by releasing IFNγ, enhancing Th1 responses [78]. Infections caused by *Leishmania* parasites elicit a swift and brief activation of NK cells. Murine models have shown that such activation relies on toll-like-receptor 9-dependent stimulation of dendritic cells (DC), leading to IL-12 production. While NK cells may not be indispensable for controlling cutaneous and visceral leishmaniasis (VL) and can display immunosuppressive functions, they play a crucial role as a source of interferon (IFN)-γ that drives antileishmanial activity in macrophages and contributes to setting up a protective T helper cell response [78].

However, it has been observed that miltefosine may negatively affect the anti-leishmanial activity of NK cells. In a study by Bulté et al. [79], the researchers evaluated the intricate interplay between *Leishmania* parasites, the host immune system, and miltefosine by generating miltefosine-resistant (MIL-R) and miltefosine-sensitive (MIL-S) *L. infantum* double-reporter strains to further characterize infection dynamics [79]. Results obtained from two different murine models revealed that MIL-R parasites triggered an elevated innate immune response characterized by increased influx and infection of neutrophils, monocytes, and dendritic cells in the liver. This response was accompanied by elevated serum IFN-γ levels, leading to a less efficient establishment in liver macrophages. The heightened IFN-γ levels originated from an increased response of hepatic NK and NKT cells to MIL-R parasites [79]. Remarkably, MIL was found to increase the in vivo fitness of MIL-R parasites by lowering NK and NKT cell activation, resulting in reduced IFN-γ production, while the attenuated phenotype of miltefosine-R parasites and their peculiar drug dependency were linked to the differential induction of innate immune responses in the liver. Overall, the results from this investigation suggest that miltefosine may select resistant parasites and lead to treatment failure [79].

Perhaps one of the most important cell populations driving *Leishmania* pathogenesis is macrophages [80]. Macrophages are at the center of disease pathogenesis. The interaction between macrophages and *Leishmania* is pivotal in the pathogenesis of the infection, serving both as a crucial environment for the survival, replication, and differentiation of parasites and as a frontline defense for their elimination. In experimental leishmaniasis models, macrophages play a fundamental role in shaping the outcome of the infection [56]. Importantly, macrophages exhibit versatility during leishmaniasis, differentiating into distinct M1 or M2 phenotypes [81].

M1 macrophages contribute to disease control by producing pro-inflammatory cytokines, NO and ROS, thereby enhancing Th1 responses [81]. This activation promotes an environment conducive to combating the infection. On the other hand, M2 macrophages play a role in disease progression by increasing the production of immunosuppressive cytokines such as IL-10 and TGFβ, supporting Th2 responses. This duality in macrophage polarization underscores their central role in orchestrating the immune response during leishmanial infection, influencing the delicate balance between parasite survival and host defense [81].

Miltefosine has been shown to exert diverse effects on macrophage lipid homeostasis [82], including its role in inhibiting the assembly of the NLRP3 inflammasome—a crucial component of the innate immune response against *Leishmania*. During infection, *Leishmania* parasites trigger the activation of the NLRP3 inflammasome to restrict intracellular parasite replication [82]. Intriguingly, miltefosine‘s influence in terms of inhibiting NLRP3 inflammasome assembly could affect the host’s ability to mount an effective defense against *Leishmania*. The intricate interplay between miltefosine, macrophage lipid dynamics, and the NLRP3 inflammasome highlights the complexity of the host–parasite interaction and underscores the potential role of miltefosine in modulating the immune response during leishmanial infection. As in the case of NK cells, the immunomodulatory effects of miltefosine on macrophages deserve to be explored more in depth, as they may hold the key to better understanding the mechanisms behind miltefosine resistance.

Collectively, the intricate immune responses in leishmaniasis, influenced by miltefosine, reveal a complex interplay with various immune cells (Figure 6). Neutrophils, eosinophils, mast cells, NK cells, and macrophages all respond differently to miltefosine, influencing the delicate balance between parasite survival and host defense. This, in addition to studies proving that miltefosine-induced activation of Th1 cytokines driven by enhanced gamma interferon (IFN-γ) and interleukin 12 (IL-12) levels prevail over Th2 responses, further confirms miltefosine’s positive immunomodulatory effect against *Leishmania* [83]. These findings highlight the drug’s potential impact on disease pathogenesis and underscore the need for further exploration to understand the mechanisms behind miltefosine resistance and the plethora of antiparasitic effects it exerts. Improved insights into these interactions could collectively pave the way for enhanced treatment strategies with this drug.

## 7. Concluding Remarks and Perspectives

This review underscores the multifaceted impact of miltefosine on trypanosomatids, including its multi-targeted effects on sphingosine-activated plasma membrane Ca^2+^ channel, mitochondrial membrane potential, and acidocalcisome alkalization. These mechanisms lead to significant intracellular Ca^2+^ accumulation, a novel finding in drug action against these parasites. While disruption of Ca^2+^ homeostasis is emerging as a therapeutic target for other protozoan parasites [84], miltefosine’s success in this regard, combined with its known perturbation of phospholipid synthesis and potent cytochrome C oxidase inhibition, contributes to parasite death, explaining its efficacy as an oral monotherapy. Moreover, miltefosine’s broad action on the host immune system, particularly in leishmaniasis, enhances its therapeutic benefits. However, the potential for miltefosine-induced resistance remains a concern. Therefore, combination therapy, such as miltefosine with amiodarone, is highly recommended to mitigate this risk, as demonstrated by the successful parasitological cure observed in mice mentioned previously.

In conclusion, further research is needed to deepen our understanding of the powerful pleiotropic effects of miltefosine and its potential as a promising drug for the widespread treatment of trypanosomatid infections. Such endeavors will not only elucidate its mechanisms of action, but also inform the development of novel therapeutic strategies to combat these neglected tropical diseases effectively.

## Figures and Tables

**Figure 1 biomolecules-14-00406-f001:**
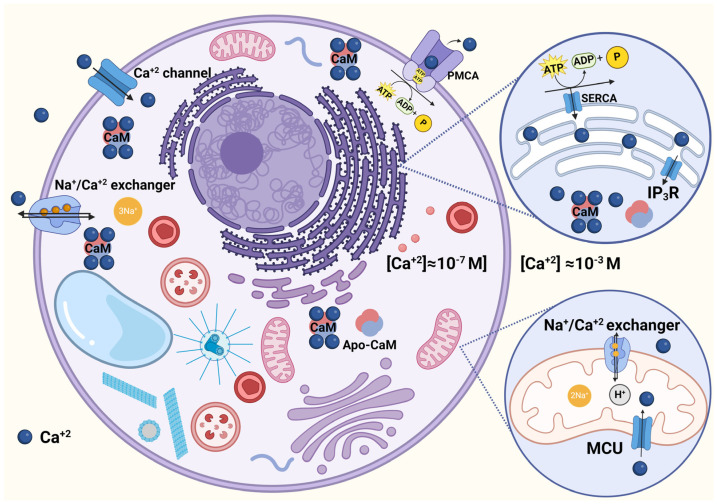
**Schematic representation of the mechanisms involved in the intracellular Ca^2+^ regulation in a mammal cell**. At the plasma membrane level, a CaM-stimulated Ca^2+^-ATPase (PMCA) responsible for Ca^2+^ extrusion and a Na^+^/Ca^2+^ exchanger can be seen. A Ca^2+^ channel for Ca^2+^ entry is also shown. One mitochondria is enlarged (right), showing the mitochondrial Ca^2+^ uniporter (MCU) for Ca^2+^ entry and a Na^+^/Ca^2+^ (or H^+^/Ca^2+^) exchanger for Ca^2+^ extrusion to the cytoplasm. The different targets for CaM regulation are also depicted. A representation of the endoplasmic reticulum (ER) is enlarged (right), with an SERCA Ca^2+^ Pump and the IP _3_ receptor. IP_3_R (Ca^2+^ channel) for Ca^2+^ release is also presented, whose regulation by CaM is highlighted (see text for details). Created with BioRender.com.

**Figure 2 biomolecules-14-00406-f002:**
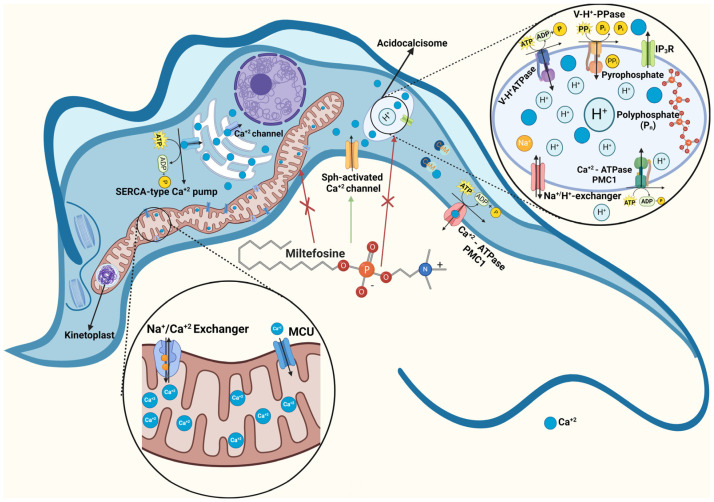
**Schematic representation of the mechanisms involved in the intracellular Ca^2+^ regulation in *Trypanosomatids*.** The unique giant mitochondrion is exposed with its kinetoplast. A detail is depicted in the inset below the main figure, showing the mitochondrial Ca^2+^ uniporter (MCU) for Ca^2+^ entry and a Na^+^/Ca^2+^ (or H^+^/Ca^2+^) exchanger for Ca^2+^ extrusion to the cytoplasm. Acidocalcisomes are enlarged at the right, depicting a PMCA-type Ca^2+^-ATPase, a Na^+^/Ca^2+^ exchanger, a V-H^+^-ATPase responsible for H^+^ accumulation, and a V-H^+^-PPase. The IP_3_ receptor, IP_3_ R (Ca^2+^ channel), for Ca^2+^ release is also presented. In the main figure, a SERCA-type Ca^2+^ pump at the endoplasmic reticulum is also depicted. At the plasma membrane level, a PMCA-type Ca^2+^-ATPase (PMC1) responsible for Ca^2+^ extrusion and the Sph-activated Ca^2+^ channel for Ca^2+^ entry is also shown. The different targets for miltefosine action are pointed out by arrows (see text for details). (Created with BioRender.com).

**Figure 3 biomolecules-14-00406-f003:**
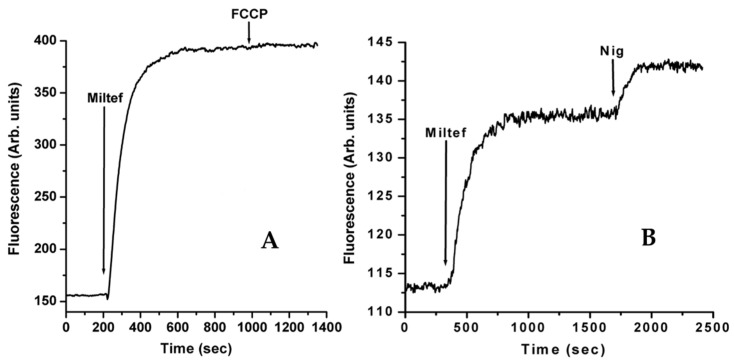
(**A**) **Effect of miltefosine on the mitochondrial electrochemical potential of *L. donovani* promastigotes.** Parasites were incubated in the presence of rhodamine 123 (10 mg/mL) for 30 min at room temperature. The release of Rhodamine 123 from the mitochondrion was followed in a fluorimeter at 29 °C under continuous stirring. The excitation wavelength was 488 nm, and emission was at 530 nm. Miltefosine (4 µM) was added (arrow), followed by FCCP (2 µM), in the presence of 2 mM of extracellular Ca^2+^. (**B**) **Effect of miltefosine on acidocalcisomes in *L. donovani*** promastigotes. Parasites were loaded with acridine orange. The release of the fluorophore was followed in a fluorimeter at 29 °C under continuous stirring. The excitation wavelength was 488 nm, and emission was at 530 nm. (**A**): Miltefosine (4 µM) was added (arrow) directly to the stirred cuvette with promastigotes loaded with acridine orange, followed by the addition of nigericin (2 µM). Data from [10]. Reproduced with permission from the American Society for Microbiology.

**Figure 4 biomolecules-14-00406-f004:**
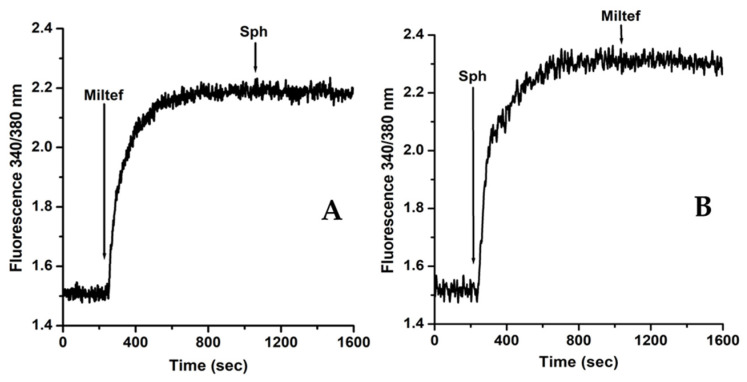
**Effect of miltefosine and sphingosine on the intracellular Ca^2+^concentration of *L. donovani* promastigotes.** Promastigotes were loaded with Fura 2 and the indicated compounds were added directly to the cuvette. (**A**) Miltefosine (4 µM) was added (arrow) in the presence of 2 mM of extracellular Ca^2+^, followed by the addition of sphingosine (10 µM). (**B**) Sphingosine (10 µM) was added as indicated (arrow), followed by miltefosine (4 µM). Data from [10]. Reproduced with permission from the American Society for Microbiology.

**Figure 5 biomolecules-14-00406-f005:**
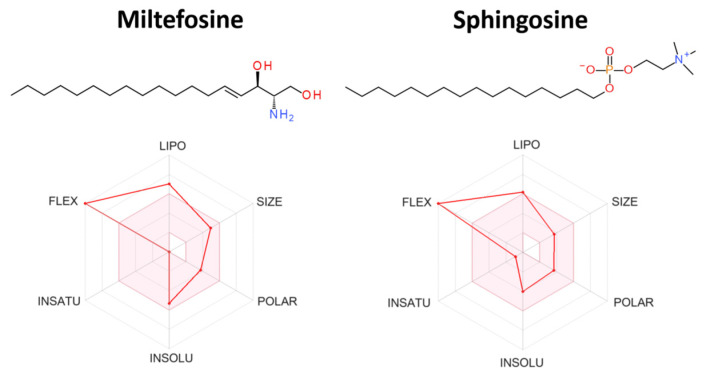
The analysis of pharmacokinetic parameters of miltefosine and sphingosine. The red area is a suitable physicochemical space for oral bioavailability obtained using the SWISSADME software [52]. Lipophilicity (LIPO): XLOGP3 between −0.7 and +5; molecular weight (SIZE): MW between 150 and 500 g/mol; polarity (POLAR) TPSA between 20 and 130 Å^2^; solubility (INSOLU): log S (ESOL) between −6 and 0; saturation (INSATU): fraction of carbons in the sp3 hybridization between 0.25 and 1; and flexibility (FLEX): no more than nine rotatable bonds (SWISSADME server [52]).

**Figure 6 biomolecules-14-00406-f006:**
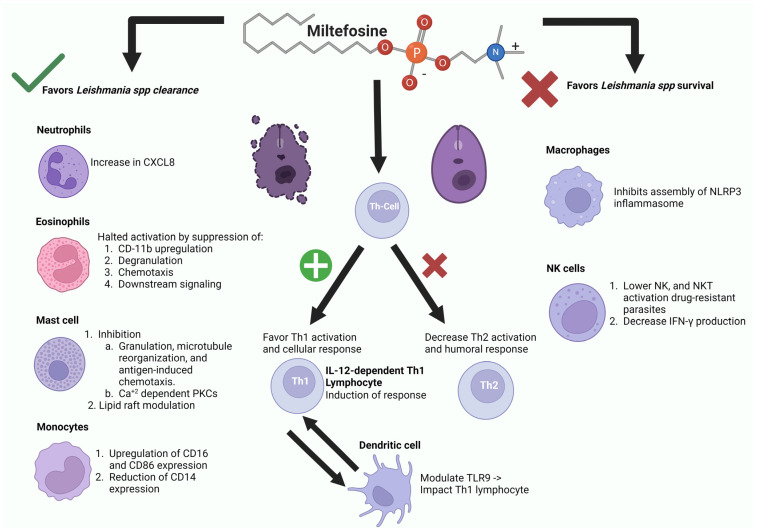
Pleiotropic immunomodulatory effects of miltefosine. Depicting positive (green) and negative (red) mechanisms influencing Leishmania parasite clearance and persistence. Note the dynamic interplay among immune cells, including neutrophils, eosinophils, mast cells, NK cells, and macrophages, in response to miltefosine. Mechanisms from the granulocytic lineage involve: (i) Neutrophils clearing parasites through phagocytosis, ROS release, NETs, and increased CXCL8 production. (ii) Eosinophil activation reducing inflammation and tissue destruction and suppressing eosinophilic effector reactions. (iii) Inhibition of mast cell degranulation, microtubule reorganization, and chemotaxis, affecting lipid raft modulation and calcium-dependent PKCs while enhancing CD16 and CD68 in monocytes. Miltefosine enhancing M1 macrophages to produce pro-inflammatory cytokines, NO, and ROS, reinforcing Th1 cytokine responses, and creating a hostile environment for parasite survival. Created with BioRender.com).

**Table 1 biomolecules-14-00406-t001:** In silico evaluation of the physicochemical properties of the compounds.

Name	LogP	MW	Hba	Hbd	Rotb	Viol	LogSw	TPSA
Miltefosine	6.79	407.57	4	0	20	0	−5.23	68.4
Sphingosine	5.28	299.49	3	3	15	0	−4.03	66.48

LogP, partition coefficient; MW, molecular weight; Hba, hydrogen bond acceptors; Hbd, hydrogen bond donors; Rotb, rotatable bonds; Viol, Lipinski’s violations; LogSw: water solubility; TPSA: Topological polar surface area.

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
