# Peer review of "Unmasking the Mechanism behind Miltefosine: Revealing the Disruption of Intracellular Ca2+ Homeostasis as a Rational Therapeutic Target in Leishmaniasis and Chagas Disease"

_biomolecules, 2024, doi:10.3390/biom14040406_

Round 1
Reviewer 1 Report
Comments and Suggestions for Authors
The authors review the mechanisms of action of miltefosine in trypanosomatid protozoan cells, with an emphasis on the mechanisms that regulate calcium concentration in these parasites. It also reviews the effects of miltefosine on cells and molecules of the immune system, and how this drug can modulate these elements in the face of Leishmania infections. The regulation of intracellular calcium levels, as well as the possibility of interacting better with organelles and other elements of trypanosomatid cells than with their counterparts in human cells, seems to be a promising mechanism of action for miltefosine. This also opens the door to the search for new molecules with this mechanism of action for new drugs against Leishmania and Trypanosoma. The topic is reviewed in great depth by the authors, which makes it a very necessary review for those working in this area.
However, some observations should be made:
Line 46: the reference cited for visceral leishmaniasis cases is from 2016, I suggest looking for a more recent reference on WHO websites or publications, for example.
Line 48: what are the "second-line drugs" cited, used as an alternative to antimonials?
Line 55 to 60: very long sentence with no reference to back it up.
Line 71: I think the word "with" was used instead of "which"
Line 75: I suggest changing "extreme" to "fatal"
Line 87: the word "paramount" was used here and in more places in the text; I suggest changing it to synonyms in some sentences to avoid repetition.
Line 129: "giant mitochondrion" - this term and the term "large mitochondrion" are used frequently in the text, I would like to know if there is any reason not to use the word "kinetoplast".
Line 153: replace "and very importantly" with "and its role"
Line 153 to 157: I suggest adding references to this passage.
Line 185: the authors use the abbreviation "CaM" for the first time, but do not describe what it is. I suggest adding the word "calmodulin" the first time the term appears.
Line 237 to 241: no reference cited.
Line 263: reference missing in square brackets.
Line 267: use of "L." instead of citing the genus Leishmania. The authors need to standardize the use of italics or not when referring to the genus Leishmania, as both forms can be found throughout the text.
Line 279 to 286: a bit confusing, I suggest rewriting. And add references, the whole passage has no citations, even though the authors say that there is a "large discussion" on the subject in line 279.
Line 309: I suggest using the expression "flow cytometry" instead of "flux cytometry".
Line 326: the word miltefosine is capitalized and lowercased throughout the text, I suggest standardizing it. The same goes for sphingosine.
Lines 330 to 333 and lines 340 to 344: the endings of the two paragraphs are very similar, it's a bit repetitive. I suggest rewriting or merging the two paragraphs into one.
Line 369: should the table legend have a different font size from the text as in the figures? I suggest checking the magazine's suggested formatting.
Lines 369 to 380: text in a different color to the rest of the text, should be standardized.
Line 383: "increasing the open probability" - I don't understand what this expression means in this sentence, perhaps it should be revised and reworded.
Line 406: "This section of the manuscript" - I suggest changing this phrase to "Here, we explore...".
Lines 405 to 417: excerpt with no reference cited.
Line 432: "thus aiding Leishmania survival and pathology", in this passage it seems that the mechanism of action cited of neutrophil activation would decrease the survival of the parasite, and not help it.
Line 487 to 490: the authors start the sentence by quoting "cytokines and immune mediators" but at the end of the sentence (line 490) they quote NK cells. Reword the beginning of the sentence.
Line 515: use of the abbreviation MIL to refer to miltefosine. This is the first time it has been used in the text, so I suggest standardizing whether you use the whole word miltefosine or its abbreviation.
Line 517: MIL-R or MIL-S? It seems that here the MIL-S strain would fit better in the context of this sentence.
Line 526 and 527: the authors repeat the definition and abbreviation of ROS and NO a few times. Check throughout the text, and only do it once.
Line 575: the references don't seem to be standardized, please check and standardize the form.
Comments on the Quality of English Language
I think the English text is fine, except for some very long sentences, or some that need to be revised, as mentioned in the comments to the authors. A medium to small revision would make the text more fluid and understandable.
Author Response
Dear Editor
We have read with interest and acknowledge all the reviewers comment. Accordingly, have included all the suggestions
Without any doubt, they will increase the overall quality of the manuscript
We recognize and appreciate very much their effort
Best wishes
Gustavo Benaim
Open Review 1
(x) I would not like to sign my review report
( ) I would like to sign my review report
Quality of English Language
( ) I am not qualified to assess the quality of English in this paper
( ) English very difficult to understand/incomprehensible
( ) Extensive editing of English language required
( ) Moderate editing of English language required
(x) Minor editing of English language required
( ) English language fine. No issues detected
Is the work a significant contribution to the field? |
|
Is the work well organized and comprehensively described? |
|
Is the work scientifically sound and not misleading? |
|
Are there appropriate and adequate references to related and previous work? |
|
Is the English used correct and readable? |
Comments and Suggestions for Authors
The authors review the mechanisms of action of miltefosine in trypanosomatid protozoan cells, with an emphasis on the mechanisms that regulate calcium concentration in these parasites. It also reviews the effects of miltefosine on cells and molecules of the immune system, and how this drug can modulate these elements in the face of Leishmania infections. The regulation of intracellular calcium levels, as well as the possibility of interacting better with organelles and other elements of trypanosomatid cells than with their counterparts in human cells, seems to be a promising mechanism of action for miltefosine. This also opens the door to the search for new molecules with this mechanism of action for new drugs against Leishmania and Trypanosoma. The topic is reviewed in great depth by the authors, which makes it a very necessary review for those working in this area.
However, some observations should be made:
Line 46: the reference cited for visceral leishmaniasis cases is from 2016, I suggest looking for a more recent reference on WHO websites or publications, for example.
We have changed the reference for a more recent one.
1.- Leishmaniasis. World Health Organization. https://www.who.int/news-room/fact-sheets/detail/leishmaniasis. Accessed: March 19, 2024.
We also actualized the data: 95 %fatality if left untreated and an estimate 50.000 to 90.000 new cases per years worldwide.
Line 48: what are the "second-line drugs" cited, used as an alternative to antimonials?
We have eliminated the phrase “as well as other second-line drugs”
Line 55 to 60: very long sentence with no reference to back it up.
We have added a couple of references to this long sentence: [3] and [4], lines 57 and 59
Line 71: I think the word "with" was used instead of "which"
The reviewer is right. It has been amended. Now line 74
Line 75: I suggest changing "extreme" to "fatal"
Changed as recommended. Now line 74
Line 87: the word "paramount" was used here and in more places in the text; I suggest changing it to synonyms in some sentences to avoid repetition.
The word paramount has been changed by the word vital. Now line 76
Line 129: "giant mitochondrion" - this term and the term "large mitochondrion" are used frequently in the text, I would like to know if there is any reason not to use the word "kinetoplast".
Actually, the kinetoplast represents the mitochondrial DNA, being is so huge in trypanosomatids that is easily visible with a light microscope, but is not properly the mitochondrion. This mitochondrion is indeed very large since it represents the 12 % of the total parasite volume.
Line 153: replace "and very importantly" with "and its role"
Replaced as suggested. Now line 152
Line 153 to 157: I suggest adding references to this passage.
A couple of references has been now added: [23,25].
Line 185: the authors use the abbreviation "CaM" for the first time, but do not describe what it is. I suggest adding the word "calmodulin" the first time the term appears.
The reviewers is right. We have amended it; calmodulin (CaM). Now line 184
Line 237 to 241: no reference cited.
2 references have now been added: [10] and [35]. Lines 236 and 238
“…cells [10] including diverse positive effects on the immune system of the infected host, apparently also responsible for its beneficial action [35].”
Line 263: reference missing in square brackets.
The reference has been added: [39]. Now line 262
Line 267: use of "L." instead of citing the genus Leishmania. The authors need to standardize the use of italics or not when referring to the genus Leishmania, as both forms can be found throughout the text.
This has now been standardized throughout the text. It now appears only Leishmania in italics (the changes are colored in red in the text now).
Line 279 to 286: a bit confusing, I suggest rewriting. And add references, the whole passage has no citations, even though the authors say that there is a "large discussion" on the subject in line 279
We have rewritten the entire sentence in an effort to clarify its meaning. Additionally, we have incorporated two references as recommended ([43-44]). Please note that we have adjusted the order of reference 44 to reference 45 to minimize major changes in the reference order. Now Lines 278 to 286
“Concerning apoptosis in unicellular organism there has been a large discussion regarding the evolution of apoptotic mechanism in individual cells, since this “altruistic” behavior is in principle not expected, since the actual view recognize this phenomenon as a mechanism of depuration of undesirable cells that would be detrimental for a multicellular individual organism as a whole [43-44]. However, taking population survival as a purpose, it is not difficult to conceive that it makes sense to expense some individual parasites in benefit of the survival of the host. This is particularly valid in case of severe infections causing the host death, and consequently conducting to the interruption of the transmission to new individuals, which indeed would be an evolutionary disadvantage. In any case, an apoptotic-like mechanism induced by miltefosine has been extensively demonstrated in several other reports in trypanosomatids, which includes a vast quantity of apoptotic parameters [43-45].”
Line 309: I suggest using the expression "flow cytometry" instead of "flux cytometry". Now line 307
This was a “typo” and now is amended.
Line 326: the word miltefosine is capitalized and lowercased throughout the text, I suggest standardizing it. The same goes for sphingosine.
We have standardized miltefosine and sphingosine in the whole document.
Lines 371, 379, 398, 403, 427, 507, 514. 544, showing the word in red
Lines 330 to 333 and lines 340 to 344: the endings of the two paragraphs are very similar, it's a bit repetitive. I suggest rewriting or merging the two paragraphs into one.
Now line 333
We agree with this reviewer suggestion. Accordingly, we have eliminated part of the first paragraph and merged the two paragraphs in one, as suggested (Changes are marked in red, lines 330-343).
Line 369: should the table legend have a different font size from the text as in the figures? I suggest checking the magazine's suggested formatting.
This have been corrected
Lines 369 to 380: text in a different color to the rest of the text, should be standardized.
This have been also corrected
Line 383: "increasing the open probability" - I don't understand what this expression means in this sentence, perhaps it should be revised and reworded.
The reviewer is right. This deserves further explanation. “Increasing the open probability” is an electrophysiological term. The function of ion channels to mediate the flux of ions through membranes of living cells depends on their number, conductance, and open probability. The open probability, PO, characterizes gating of channels that is sensitive to experimental conditions and that can be determined in single-channel experiments.
Since this term is not easy to explain and it is of very limited contribution to the scope of this review, we have eliminated it.
Line 406: "This section of the manuscript" - I suggest changing this phrase to "Here, we explore...".
Changed as suggested. Now line 405
Lines 405 to 417: excerpt with no reference cited.
Lines 405 to 417 serve as an introductory opening for this section and do not contain citable content. The intention is to outline what will be discussed in the following lines.
Top of Form
Line 432: "thus aiding Leishmania survival and pathology", in this passage it seems that the mechanism of action cited of neutrophil activation would decrease the survival of the parasite, and not help it.
Correct. This statement aims to emphasize contrasting results in the balance between survival and containment of infection and should be interpreted in the context of the preceding and following studies cited herein.
Line 487 to 490: the authors start the sentence by quoting "cytokines and immune mediators" but at the end of the sentence (line 490) they quote NK cells. Reword the beginning of the sentence.
The reviewer is right This has been corrected. The sentence has been reworded and now reads: ‘Various cells, cytokines and immune mediators’. Now lines 484-485
Line 515: use of the abbreviation MIL to refer to miltefosine. This is the first time it has been used in the text, so I suggest standardizing whether you use the whole word miltefosine or its abbreviation.
Done. It now reads: “Miltefosine-R parasites”. Now line 513
Line 517: MIL-R or MIL-S? It seems that here the MIL-S strain would fit better in the context of this sentence.
No. This sentence does not refer to specific parasite strain but rather to the overall phenomenon of selection for resistance.
Line 526 and 527: the authors repeat the definition and abbreviation of ROS and NO a few times. Check throughout the text, and only do it once.
Done. Now lines 522 and 524
Line 575: the references don't seem to be standardized, please check and standardize the form.
The reviewer is right. All the references are now standardized according to the journal guidelines.
Comments on the Quality of English Language
I think the English text is fine, except for some very long sentences, or some that need to be revised, as mentioned in the comments to the authors. A medium to small revision would make the text more fluid and understandable.
We accepted this suggestion, and accordingly have revised the whole manuscript and introduced some changes, as are denoted in the precedent lines.
Reviewer 2 Report
Comments and Suggestions for Authors
The manuscript by Benaim & Paniz-Mondolfi provides a comprehensive review of the currently known mechanisms of action of Miltefosine, a commonly used drug for the treatment of Leishmaniasis and Chagas disease. In this work, the authors highlight the direct effect of miltefosine on calcium homeostasis and the host immune system, guiding the reader trough the differences between two clinically important parasite species: Trypanosome and Leishmania. Moreover, the authors review the different targets of Miltefosine within the parasite, elucidating the relationship between target and effect.
Overall, the manuscript is well-structured and covers a wide range of published articles in the field. Additionally, the focus of the article is quite interesting for readers in the field. However, certain adjustments and clarifications are required to meet the standards of publication in Biomolecules.
Specific comments.
1. Despite the introduction and the effects of Miltefosine on the host immune system sections being well-written and structured, I found difficult to follow the ideas presented in the mechanism of action of miltefosine in trypanosomatids section. Therefore, I kindly request that the authors consider revising the English language in the manuscript to enhance clarity and readability.
2. The authors should update the information regarding the number of cases of visceral leishmaniasis worldwide (lines 44-46). According to the World Health Organization, the annual number of cases of this disease ranges between 50,000 to 90,000. Please correct this information accordingly.
3. In general the citations are well presented, however some mistakes should be addressed:
a. The reference #3 does not appear in the main text.
b. In line 263 there is a pair of square brackets but no reference inside them.
4. Line 68: change ‘its’ for ‘their’ as you are referring to plural parasites.
5. Line 150: it appears that there is a redundancy in the phrase ‘when transformed to blood stages, circulating parasites in the blood the osmolarity returns to…’, I suggest to remove the part ‘circulating parasites in the blood’.
6. Line 202: there is a missing period in T. equiperdum
7. As mentioned before, it was quite difficult to follow the ideas in the mechanism of action of miltefosine in trypanosomatids section due to several grammatical and orthographic errors. For instance, in line 244 'modifies' should be 'modified'; in line 267, the species of Leishmania is missing; in line 269, the authors mention a dramatic 5-fold increase but fail to specify what increased; in line 270, remove the last 's' from 'increase'; in line 270, 'were' should be 'was', etc. Moreover, the section is composed of long phrases. The authors should consider shortening these phrases and highlighting the important information contained within them."
8. Line 309 should be flow cytometry instead of flux cytometry, and separate the- from more.
9. Lines 313 to 316 and 334-335: the ideas are difficult to understand, please re-phrase.
10. Include the citation in Figure 3. Moreover, it seems to be a mistake that the authors describe (A) in figure 3B; in any case, it would be better to merge the titles of 3A and 3B and describe the two conditions as in Figure 4.
11. In the section Effects of Miltefosine on the host immune system, the genus Leishmania should always be presented with italics, please correct it in lines 408, 414, 419, 439, 445, 459, 468, 475, and 476. Also, two periods are missing in lines 448 and 452 just after the citations.
Author Response
Open Review 2
(x) I would not like to sign my review report
( ) I would like to sign my review report
Quality of English Language
(x) I am not qualified to assess the quality of English in this paper
( ) English very difficult to understand/incomprehensible
( ) Extensive editing of English language required
( ) Moderate editing of English language required
( ) Minor editing of English language required
( ) English language fine. No issues detected
Is the work a significant contribution to the field? |
|
Is the work well organized and comprehensively described? |
|
Is the work scientifically sound and not misleading? |
|
Are there appropriate and adequate references to related and previous work? |
|
Is the English used correct and readable? |
Comments and Suggestions for Authors
The manuscript by Benaim & Paniz-Mondolfi provides a comprehensive review of the currently known mechanisms of action of Miltefosine, a commonly used drug for the treatment of Leishmaniasis and Chagas disease. In this work, the authors highlight the direct effect of miltefosine on calcium homeostasis and the host immune system, guiding the reader trough the differences between two clinically important parasite species: Trypanosome and Leishmania. Moreover, the authors review the different targets of Miltefosine within the parasite, elucidating the relationship between target and effect.
Overall, the manuscript is well-structured and covers a wide range of published articles in the field. Additionally, the focus of the article is quite interesting for readers in the field. However, certain adjustments and clarifications are required to meet the standards of publication in Biomolecules.
Specific comments.
- Despite the introduction and the effects of Miltefosine on the host immune system sections being well-written and structured, I found difficult to follow the ideas presented in the mechanism of action of miltefosine in trypanosomatids section. Therefore, I kindly request that the authors consider revising the English language in the manuscript to enhance clarity and readability.
We accepted this suggestion, and accordingly we have revised the whole manuscript and introduced some changes as marked in the text in red.
- The authors should update the information regarding the number of cases of visceral leishmaniasis worldwide (lines 44-46). According to the World Health Organization, the annual number of cases of this disease ranges between 50,000 to 90,000. Please correct this information accordingly.
We have changed the reference for a more recent one.
1.- Leishmaniasis. World Health Organization. https://www.who.int/news-room/fact-sheets/detail/leishmaniasis. Accessed: March 19, 2024.
We also actualized the data: 95 %fatality if left untreated and an estimate 50.000 to 90.000 new cases per years worldwide.
- In general the citations are well presented, however some mistakes should be addressed:
- The reference #3 does not appear in the main text.
This reference now has been added in the text.
- In line 263 there is a pair of square brackets but no reference inside them.
This reference has now been added in the brackets
- Line 68: change ‘its’ for ‘their’ as you are referring to plural parasites.
Changed, as suggested
- Line 150: it appears that there is a redundancy in the phrase ‘when transformed to blood stages, circulating parasites in the blood the osmolarity returns to…’, I suggest to remove the part ‘circulating parasites in the blood’.
This part has been eliminated “circulating parasites in the blood”
- Line 202: there is a missing period in T. equiperdum
The period has been added now
- As mentioned before, it was quite difficult to follow the ideas in the mechanism of action of miltefosine in trypanosomatids section due to several grammatical and orthographic errors. For instance, in line 244 'modifies' should be 'modified'; in line 267, the species of Leishmania is missing; in line 269, the authors mention a dramatic 5-fold increase but fail to specify what increased; in line 270, remove the last 's' from 'increase'; in line 270, 'were' should be 'was', etc. Moreover, the section is composed of long phrases. The authors should consider shortening these phrases and highlighting the important information contained within them."
All the grammatical error has been corrected. The whole description of the mechanism of action of miltefosine in parasites has been revised thoroughly and modified accordingly.
- Line 309 should be flow cytometry instead of flux cytometry, and separate the- from more.
Both errors had been amended
- Lines 313 to 316 and 334-335: the ideas are difficult to understand, please re-phrase.
The paragraphs have been re-phrased
Lines 312-315
“This is the case of the antituberculosis drug SQ109 that has been shown to produce an effect in the mitochondrion of other trypanosomatids, as in the case of T. cruzi. The experiments showed a rapid collapse of the inner mitochondrial membrane potential Δφm, dissipating the proton gradient by acting as the uncoupler or protonophore FCCP [6]. The same result has been found in L. mexicana [52], and also in L. donovani [7].”
Lines 334-336
“In regard to the release of Ca2+ from the mitochondrion, since the driving force for the Ca2+ entrance is the mitochondrial membrane potential Δφm , the collapse caused by the drug action produced the release of the accumulated Ca2+ from the mitochondrion to the cytoplasm”
- Include the citation in Figure 3. Moreover, it seems to be a mistake that the authors describe (A) in figure 3B; in any case, it would be better to merge the titles of 3A and 3B and describe the two conditions as in Figure 4.
Actually, Fig. 3A and 3B albeit appearing similar figures, they represent very different experiments. Fig. 3A represents the effect of miltefosine on mitochondria while Fig.3B represents the effect of miltefosine on acidocalcisomes. We have fused the legend in one block. Thus, is not necessary to include the citation in Fig. 3A, since is the same for both figures.
- In the section Effects of Miltefosine on the host immune system, the genus Leishmania should always be presented with italics, please correct it in lines 408, 414, 419, 439, 445, 459, 468, 475, and 476.
This has now been standardized throughout the whole text. It now appears only Leishmania in italics (the changed words are colored in red in the text now).
Also, two periods are missing in lines 448 and 452 just after the citations.
We found the missing period of line 452, but no the from line 448